# Specificity of Anti-Citrullinated Protein Antibodies in Rheumatoid Arthritis

**DOI:** 10.3390/antib8020037

**Published:** 2019-06-07

**Authors:** Nicole H. Trier, Bettina E. Holm, Paul R. Hansen, Ole Slot, Henning Locht, Gunnar Houen

**Affiliations:** 1Department of Biomarkers and Autoimmunity, Statens Serum Institut, Artillerivej 5, 2300 Copenhagen S, Denmark; 2Department of Clinical Immunology, Rigshospitalet, Ole Maaløes vej 26, 2200 Copenhagen N, Denmark; Bettina.eide.holm@regionh.dk; 3Department of Drug Design and Pharmacology, Universitetsparken 2, 2100 Copenhagen Ø, Denmark; prh@sund.ku.dk; 4Department of Rheumatology, Rigshospitalet Glostrup, Nordre Ringvej 57, 2600 Glostrup, Denmark; ole.slot.01@regionh.dk; 5Department of Rheumatology, Frederiksberg Hospital, Nordre Fasanvej 57, 2000 Frederiksberg, Denmark; Henning.locht@regionh.dk

**Keywords:** anti-citrullinated protein antibodies, citrulline, homo-citrulline, epitopes, peptides, rheumatoid arthritis

## Abstract

Rheumatoid arthritis (RA) is an autoimmune disease of unknown etiology. The majority of individuals with RA are positive for the disease-specific anti-citrullinated protein antibodies (ACPAs). These antibodies are primarily of cross-reactive nature, hence, the true autoantigen to ACPA remains unidentified. In this study, we analyzed the reactivity of RA sera to several post-translationally modified epitopes, in order to further characterize the specific nature of ACPAs by immunoassays. Substituting citrulline with other amino acids, e.g., D-citrulline, homo-citrulline and methyl-arginine illustrated that ACPAs are utmost specific for citrullinated targets. Collectively, these findings support that ACPAs and citrullinated targets are specific for RA, making citrulline-containing peptide targets the most effective assays for detection of ACPAs.

## 1. Introduction

Rheumatoid arthritis (RA) is an autoimmune disease of unknown etiology, which causes inflammation in the joints and in severe cases erosion of the underlying bone [1]. The disease primarily affects women with a 2:1 female/male ratio and onset of disease symptoms typically peak in the forties to sixties [2]. The disease affects approximately 1% of the population worldwide [1,3]. 

Several autoantibodies are associated with RA. One of the most known is the rheumatoid factor (RF), which recognizes the fragment crystallizable (Fc) domain of IgGs, that is the second and third constant regions of IgG. Although present in 60%–70% of individuals with RA, this group of antibodies is not specific for RA as RF also occasionally is detected in other connective tissue diseases [4,5]. 

Another group of antibodies that is associated with RA is the anti-citrullinated protein antibodies, also referred to as ACPAs. ACPAs recognize the post-translationally modified amino acid citrulline, originating from the amino acid arginine [6,7]. The peptidylarginine deiminase (PAD) catalyzes the conversion of arginine to citrulline, where the nitrogen atom is replaced with an oxygen atom and the positive charge of the guanidino group is eliminated [8]. 

ACPAs are detected in up to 80% of serum from RA individuals and have been reported to be associated with a more severe disease course and disease outcome compared to ACPA-negative RA, and may even precede disease symptoms [9,10,11,12]. These antibodies are specific for RA, although they also have been sporadically reported to be associated with other diseases [13].

Several antibody-based assays exist for detection of ACPAs. The majority of the assays employ cyclic citrullinated peptides (CCP) [12,14,15,16]. Dependent on the origin of the assays and the number of peptides, the assays are divided into three generations, CCP1–CCP3 [9,12,14,15,16]. The assays employ different peptides, indicating that the presence of citrulline, rather than a specific citrullinated epitope itself, is essential for ACPA detection, which is in accordance with their cross-reactive nature. ACPAs recognize several citrullinated protein targets, preferably with a Cit–Gly motif [7,17,18,19,20,21]. Although no true citrullinated autoantigen has been identified, several citrullinated targets have been reported, e.g., collagen, fibrinogen, Epstein-Barr nuclear antigen (EBNA)-1, EBNA-2, α-enolase, and vimentin [18,22,23,24,25,26,27]. As a consequence, ACPAs have been described as a group of antibodies with overlapping reactivity, characterized by significant cross-reactivity to citrullinated targets [24,25,28,29].

Several studies have analyzed the importance of the amino acids surrounding citrulline for a stable antibody–antigen interaction. Studies find that the Cit–Gly motif is essential for antibody reactivity, although other motifs occasionally are tolerated as well [17,20,21]. Substitution studies illustrate that substitutions in positions –x–x–Cit–G–x– have no influence on antibody reactivity, clearly illustrating the importance of the central Cit–Gly motif rather than a specific epitope, which the majority of antigens demonstrate [17,19,20,21,23]. However, only limited information is available in relation to modification of the specific amino acid citrulline.

It has been reported that ACPAs occasionally may interact with homo-citrullinated peptides as well [30]. Homo-citrulline is similar to citrulline, a post-translational modification of arginine. The functional group of homo-citrulline and citrulline is identical, but the carbon backbone of homo-citrulline contains and additional -CH2- compared to citrulline. Antibodies recognizing citrulline- and homo-citrulline-containing peptides contain partly overlapping binding sites in RA sera, suggesting some kind of relationship between these two groups of antibodies [31,32,33]. The relationship between homo-citrulline and RA remains unclear.

In the present study, we examined the importance of the specific citrulline unit in order to obtain further knowledge about the nature of citrulline-dependent antibody–antigen interactions and ACPA response in RA patients. This was done by substituting citrulline with amino acids of similar functionality, e.g., D-citrulline, homo-citrulline, methyl-arginine, and arginine, and analyzing antibody reactivity in traditional immunoassays. We found that antibody reactivity was significantly reduced when citrulline was substituted with D-citrulline, homo-citrulline, or methyl arginine. These findings illustrate that the presence of L-citrulline is crucial for specific antibody reactivity, and even the smallest alterations in the amino acid side chain or side chain presentation interferes with the specific antibody–antigen interaction.

## 2. Materials and Methods

### 2.1. Materials

Synthetic citrulline-, arginine- and homo-citrulline-containing peptides were from Schafer-N (Lyngby, Denmark). Streptavidin, alkaline phosphatase (AP)-conjugated goat anti-human IgG and *para*-nitrophenylphosphate (*p*NPP) were from Sigma Aldrich (St. Louis, Mo, USA). Tris–Tween–NaCl (TTN) buffer (0.05 M Tris, 0.3 M NaCl, 1% Tween 20, pH 7.4) and AP-substrate buffer (1 M diethanolamine, 0.5 mM MgCl_2_, pH 9.8) were from SSI Diagnostica (Hillerød, Denmark).

#### 2.1.1. Patient Sera

Twenty-five RA-positive sera, diagnosed according to the American College of Rheumatology (ACR) classification criteria [34] were enrolled in this study. The sera were obtained from Rigshospitalet Glostrup. The project was approved by the scientific ethics committees in Denmark (Project ID: 19980024 PMC and H-15009640). Healthy control (HC) sera were obtained from volunteers at Rigshospitalet and at Statens Serum Institut (Copenhagen, Denmark). All participating subjects gave their written informed consent for inclusion before they participated in the study. Twenty-five RA sera were enrolled of which 22 were from females. The average age for RA patients was 55 years. All RA sera were CCP2 positive, with CCP titers ranging from 25 to 3200 U/mL. Eighteen RA sera were RF IgA positive (>15 U/mL) and 21 were RF IgM positive (15 IU/mL). Three sera were RF negative.

#### 2.1.2. Synthetic Peptides

Synthetic peptides containing methylarginie were synthesized by traditional fluorenylmethyloxycarbony peptide synthesis, as previously described [35]. The peptides were approximately 20 amino acids long with a central Cit/hCit/Arg (*N*^ω−^Me)/Arg residue. The modified amino acids introduced are presented in Figure 1.

As presented in Table 1, the peptides selected for analysis in this study were a combination of human proteins and viral proteins, which have previously been identified as potential ACPA targets. In particular, the citrullinated viral EBNA-2 peptide and the human pro-filaggrin peptide have been reported to be good ACPA substrates [7,12,24,25]. Thus, ACPA reactivity was analyzed based on reactivity to human peptides and cross-reactivity to viral peptides.

### 2.2. Streptavidin Capture Enzyme-Linked Immunosorbent Assay

MaxiSorp 96-well microtiter plates (Nunc, Roskilde, Denmark) were precoated with streptavidin (1 µg/mL) for 2 h at room temperature (RT) followed by coating with biotinylated peptides (1µg/mL) for 2 h at RT. Sera diluted (1:200) in TTN were added to the microtiter plates and incubated for 1 h at RT. Patient sera were analyzed in duplicates. After careful washing with TTN buffer, AP-conjugated goat anti-human IgG diluted in TTN (1 µg/mL) was added to the wells and the plates incubated for 1 h at RT. For antibody quantification, AP activity was determined with *p*NPP (1 mg/mL) diluted in AP substrate buffer. The absorbance was measured at 405 nm, with background subtraction at 650 nm, using a ThermoMax microtiter plate reader (Molecular Devices, Menlo Park, CA, USA).

### 2.3. Statistical Analyses

Statistical calculations were performed using two measurements of 25 RA sera. The values obtained in this study were compared further by using the two-tailed Student’s *t*-test for single column analysis and Mann–Whitney U-test, which compared all columns to control columns.

## 3. Results

### 3.1. Reactivity of Rheumatoid Arthritis Sera to D/L-Citrulline-Containing Peptides

To determine whether the orientation of citrulline in the epitope is essential for antibody reactivity, the reactivity of ACPA-positive RA sera was analyzed to two peptides (SHQEST-Cit-GRSRGRS), originating from pro-filaggrin by streptavidin capture enzyme-linked immunosorbent assay (ELISA). The pro-filaggrin peptide, SHQEST-R-GRSRGRS, was selected for preliminary analysis, as a peptide containing this central region originally was used for detection of ACPAs. One peptide version contained a traditional L-citrulline, whereas the other contained D-citrulline. In total, 25 RA and 20 HC sera were analyzed for reactivity.

As presented in Figure 2, the ACPA-positive RA sera reacted with the citrullinated pro-filaggrin peptides (Figure 2a).

The RA sera reacted with the D-citrulline-containing peptide as well, however, antibody reactivity was significantly reduced when replacing L-citrulline with D-citrulline (*p* < 0.0001). None of the HC sera reacted significantly with the pro-filaggrin peptides (Figure 2b). These findings indicate that the orientation of the citrulline side chain is crucial for antibody reactivity.

### 3.2. Reactivity of Rheumatoid Arthritis Sera to Citrullinated and Homo-Citrullinated Peptides

Next, the reactivity of RA sera to various citrullinated and homo-citrullinated epitopes was analyzed by streptavidin capture ELISA. In total, 10 RA sera and 10 HC sera were analyzed for reactivity. Peptides originating from pro-filaggrin and EBNA1 were selected as templates, as Epstein–Barr virus (EBV) has been proposed to be involved in the onset of RA, and because ACPAs are very cross-reactive. Some of the EBV peptides have been described as good ACPA candidates, whereas others have been described as poor candidates. This was done in order to determine whether addition of homo-citrulline would increase antibody reactivity, and ultimately, to determine whether antibody reactivity to citrullinated peptides and homo-citrullinated peptides differ from each other.

Figure 3 illustrates the reactivity of RA and HC sera to the substituted peptides.

As presented in Figure 3, significant antibody reactivity occurred with the citrullinated peptides, compared to the homo-citrullinated peptides (Figure 3a,b,e). The RA sera reacted primarily with peptide 3 and 6 of the homo-citrullinated analogues (Figure 3b) and peptides 1, 2, 3, 4 and 6 of the citrullinated analogues (Figure 3a). No reactivity was found to peptide 7, which most likely is due to the absence of a Gly residue on the C-terminal side of citrulline. The RA sera reacted weakly with the homo-citrullinated version of peptide 6, and not with the citrullinated version, which most likely is due to the presence of negatively charged amino acids, which previously have been found to influence antibody reactivity negatively, when located close to the citrulline residue [17,20]. None of the HC sera reacted with the substituted peptides (Figure 3c,d). These findings indicate that antibodies to homo-citrullinated peptides have the same restrictions as citrullinated peptides.

### 3.3. Reactivity of Rheumatoid Arthritis Sera to Arg(N^w^Me) Peptides

Methylation is another posttranslational modification of Arg, which theoretically could be relevant in relation to autoantibody reactivity. To determine whether Arg(Me) peptides were recognized by RA sera, selected Arg(Me)-, Arg- and citrulline-containing peptides, which previously have been identified as potential ACPA substrates, were screened for antibody reactivity by streptavidin capture ELISA. In total, 20 RA sera were screened for reactivity. Arg–Gly-containing peptides were used as negative control.

Figure 4 illustrates the reactivity of RA sera to peptides containing either citrulline, Arg(Me) or arginine. The RA sera reacted significantly with the citrullinated analogues compared to the arginine-containing control peptides (*p* < 0.0001). RA reactivity was found in relation to the Arg(Me) EBNA-1 (Figure 3a) and EBNA-2 peptides (3d), however, reactivity not significant compared to the controls (Arg-containing peptides). Increased levels of antibody reactivity to Arg(Me)-containing EBNA may be ascribed to the presence of EBV-specific antibodies. As can be seen, antibody reactivity was significantly reduced when replacing citrulline with Arg(Me). These findings confirm that the presence of citrulline is extremely specific and crucial for a stable ACPA interaction.

## 4. Discussion

Currently, no true autoantigens of RA have been identified, which is attributed to ACPA responses in RA sera being cross-reactive [20,24,25,28,29]. Based on this knowledge, we, like others, selected viral peptides in combination with human-derived peptides for characterization of the ACPA response, because EBV has been proposed to be involved in the onset of RA. Based on current knowledge and the fact that none of these peptides have been targeted in regular EBV immune responses [25,36], findings indicate that viral citrullinated peptides are representative of characterization of ACPAs. This is in accordance with our recent findings where we described that a single viral peptide originating from EBNA2 is just as effective in detecting ACPAs as the currently applied CCP assays, and more effective as the originally described pro-filaggrin peptide [12,24].

Antibody reactivity to D-citrulline-, L-citrulline-, homo-citrulline-, and Arg(Me)-containing peptides was determined. As can be seen, substitution of L-Cit to D-Cit significantly reduced antibody reactivity (Figure 2), indicating that a correct presentation of the side chain is crucial for a stable antibody-antigen interaction, which is accordance to several crystal structures of antibody-Cit peptide complexes [26,37].

A similar pattern applies to experiments analyzing antibody reactivity to citrulline and homo-citrullinated peptides (Figure 3). Antibody reactivity was significantly reduced to the homo-citrullinated peptides compared to the citrullinated peptides, most likely because the additional -CH_2_- unit in the side chain of homo-citrulline projects the crucial oxygen atom, making it impossible to generate the crucial hydrogen bond between the antibody and the antigen. This remains to be verified, by X-ray crystallography. A few citrullinated and homo-citrullinated peptides were not recognized by the RA sera, which most likely relates to the presence of a negatively charged amino acid close to (homo-)citrulline (GNGLGE–Xxx–GDTSGPEGSGGSG) (or the absence of a positive charge in position 3–5 C-terminal to citrulline) and the absence of a Cit–Gly motif (FAEVLKDAI–Xxx–DLVMTKPAPT). These findings indicate that antibodies recognizing homo-citrullinated and citrullinated epitopes have similar reactivity patterns (or limitations), as previously suggested [33].

It has previously been reported that the mere presence of a peptide backbone and a Cit–Gly motif is sufficient for antibody recognition [19]. Nevertheless, this most likely only applies if the side chain of citrulline is projected into the binding groove of the antibody [26]. Thorough studies performed by Uysal and colleagues characterized the interaction between a monoclonal antibody directed to a citrullinated collagen peptide (A–Cit–GLTGRPGDA) [26]. Structure analysis showed that the citrullinated epitope changes conformation upon interaction with the antibody by adopting a non-native β-hairpin conformation, where the side chain of citrulline is exposed into the combining site of the antibody [26]. This allows the peptide to have more interactions between the antibody’s complementarity-determining regions (CDR) and the amino acids side chains of the antigenic peptide, which contributes to an increased affinity. Moreover, the specific contact with the citrulline residue (through the crucial oxygen atom) illustrates that an arginine in the same position would not make contact [26]. This study has been confirmed by a very recent study conducted by Ge and colleagues, where the crystal structure of an antibody-citrullinated peptide complex was analyzed [37]. They found that the central Cit–Gly motif in combination with a flexible peptide structure was essential for antibody reactivity, as only the central motif mediated direct contact with the antibody. Based on these findings, it was concluded that citrulline specificity can be explained by direct interactions with the antibody. This model may explain why antibody reactivity is lost when substituting L-citrulline with D-citrulline. The L-citrulline residue protrudes into the CDRs, making direct contact, however, the side chain of D-citrulline protrudes away from the CDRs, increasing the distance between the interactive amino acids in the antibody and antigen. As a consequence, the amino acids are too far apart to generate a stable interaction. This remains to be verified.

The origin and biological function of citrullinated antigens in RA are unknown. Citrullination is catalyzed by PAD in a process where the positively charged guanidine group is replaced with the neutrally charged ureido group by an overall N to O exchange reaction [8]. In RA, Arg-containing peptides are candidate antigens for generation of ACPAs. Citrullination arises as a posttranslational modification in relation to infection or inflammation, and it may be regarded as a cellular response to an altered cellular state. Likewise, other Arg modifications could possibly arise in response to the altered cellular state, and methylations are a relatively common modification of Arg, which could play a role in the development of autoimmunity. Based on this knowledge, antibody reactivity to methylated Arg peptides was also analyzed. However, as presented in Figure 4, no significant antibody reactivity was found to the Arg(Me)-substituted peptides. This makes sense, as the crucial oxygen atom in the side chain of Arg is replaced with nitrogen, which does not contribute with an ionic bond but a hydrogen bond. Not even the presence of an additional CH_3_–, which may contribute with additional Van der Waals interactions may compensate for the displacement of the crucial oxygen atom.

Collectively, these findings illustrate the specific nature of ACPA reactivity to citrullinated targets. When altering the orientation of citrulline in the binding pocket of the antibody, the antibody reactivity is significantly reduced, clearly illustrating the “key–lock” approach for antibodies. No modifications are tolerated, illustrating how fragile, yet strong and specific, antibody–antigen interactions in ACPA responses are.

## Figures and Tables

**Figure 1 antibodies-08-00037-f001:**
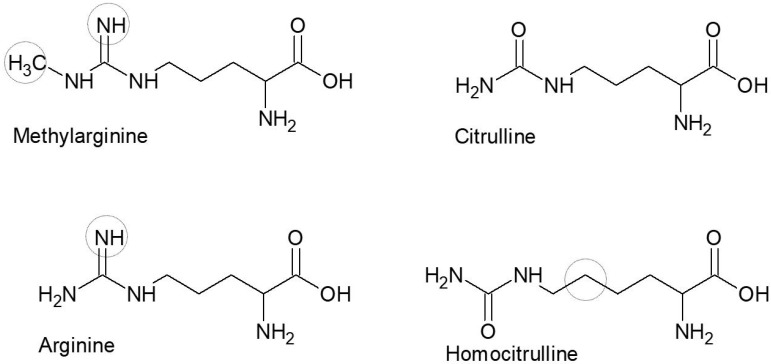
Schematic illustration of the central amino acid substitutions found in the peptides used in this study. The point of origin is citrulline; the circles represent atoms different from citrulline.

**Figure 2 antibodies-08-00037-f002:**
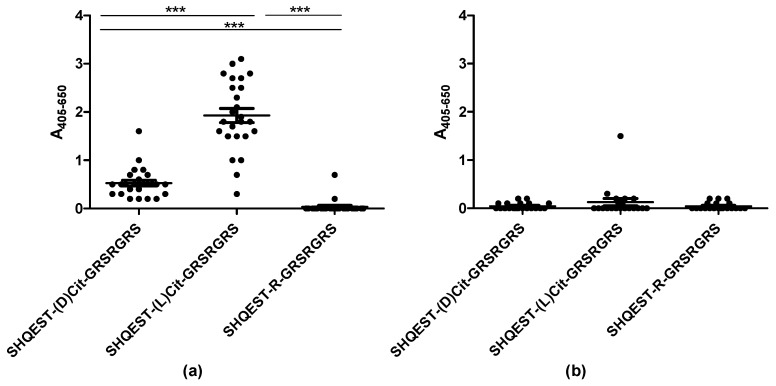
Antibody reactivity to a citrullinated pro-filaggrin peptide containing D- or L-citrulline analyzed by streptavidin capture ELISA. The Arg-containing pro-filaggrin, with a natural L-Arg was used as negative control. (**a**) Reactivity of RA sera (*n* = 25); (**b**) Reactivity of healthy control (HC) sera (*n* = 20).

**Figure 3 antibodies-08-00037-f003:**
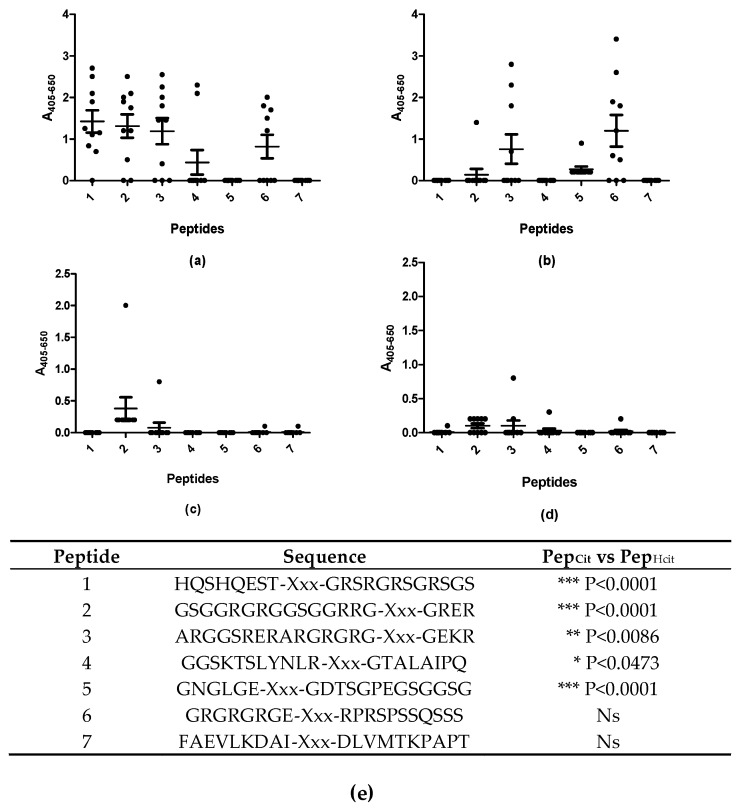
Reactivity of rheumatoid arthritis (*n* = 10) and healthy control sera (*n* = 10) to substituted peptides analyzed by streptavidin capture ELISA. (**a**) Reactivity of RA sera to citrullinated peptides; (**b**) Reactivity of RA sera to homo-citrullinated peptides; (**c**) Reactivity of HC sera to citrullinated peptides; (**d**) Reactivity of HC sera to homo-citrullinated peptides; (**e**) Peptides screened for analysis, with the statistical difference in antibody reactivity between the individual citrullinated and homo-citrullinated peptides. Xxx represent the location of the substituted amino acid in the peptide. Peptide 1: pro-filaggrin (aa 306–319). Peptide 2–7: EBNA-1 derived peptides (see Table 1 for further information). All modifications replaced the positively charged amino acid arginine.

**Figure 4 antibodies-08-00037-f004:**
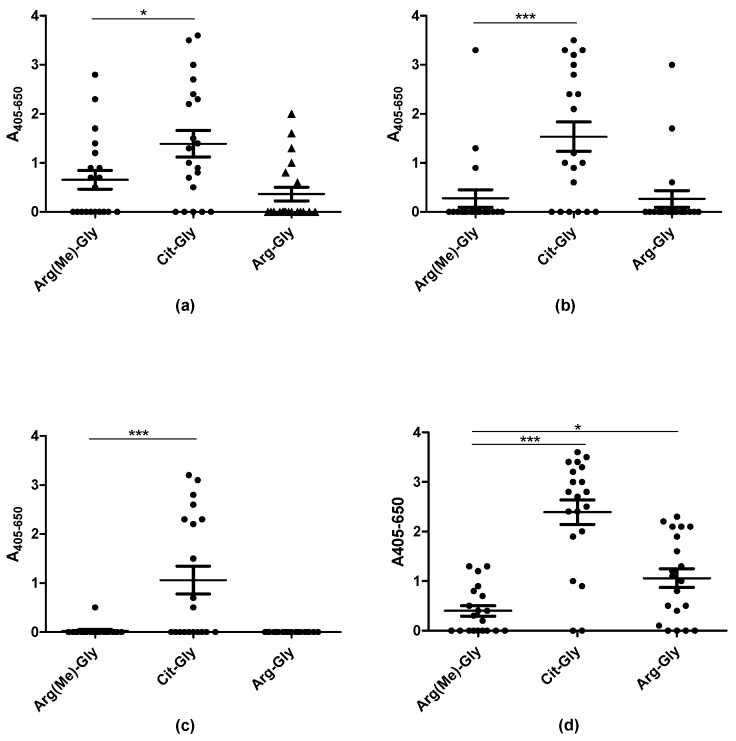
Reactivity of rheumatoid arthritis sera (n = 20) to citrullinated and Arg(Me)-substituted peptides analyzed by streptavidin capture ELISA. (**a**) Antibody reactivity to substituted peptides, using the EBNA1 peptide ARGGSRERARGRGRG–R–GEKR as template; (**b**) Antibody reactivity to substituted peptides, using the EBNA1 peptide GGSKTSLYNLR–R–GTALAIPQ as template; (**c**) Antibody reactivity to substituted peptides, using the proteoglycan peptide PQASVPLRLT–R–GSRAPISRAQ as template; (**d**) Antibody reactivity to substituted peptides using the EBNA2 peptide GQGRGRWRG–R–GRSKGRGRMH as template.

**Table 1 antibodies-08-00037-t001:** Peptides used in the current study. The underscored amino acids represent the substituted amino acid.

Peptide	aa	Origin	Substitution
SHQEST–R–GRSRGRS	306–319	Profilaggrin	Cit
HQSHQEST–R–GRSRGRSGRSGS	304–324	Profilaggrin	Cit/hCit
GSGGRGRGGSGGRRG–R–GRER	341–360	EBNA1	Cit/hCit
ARGGSRERARGRGRG–R–GEKR	361–380	EBNA1	Cit/hCit/Arg(Me)
GNGLGE–R–GDTSGPEGSGGSG	11–30	EBNA1	Cit/hCit
GRGRGRGE–R–RPRSPSSQSSS	371–390	EBNA1	Cit/hCit
FAEVLKDAI–R–DLVMTKPAPT	571–590	EBNA1	Cit/hCit
GGSKTSLYNLR–R–GTALAIPQ	511–530	EBNA1	Cit/hCit/Arg(Me)
PQASVPLRLT–R–GSRAPISRAQ	1835–1855	Proteoglycan	Arg(Me)
GQGRGRWRG–R–GRSKGRGRMH	313–332	EBNA-2	Arg(Me)

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
