# Peer review of "Specificity of Anti-Citrullinated Protein Antibodies in Rheumatoid Arthritis"

_2073-4468, 2019, doi:10.3390/antib8020037_

Reviewer 1 Report

There has lately been an increasing interest of RA autoreactivity to other modified residues besides citrulline, including carbamylation, and the manuscript contributes to that literature and our understanding of ACPA. It is well written and focus on a very interesting topic, yet there are some questions to address that would substantially improve the manuscript:

 -It is overall difficult to follow what peptides that has been used in the different figures and what protein they originate from. In Figure 3 c, what proteins do the peptides originate from? What amino acid residues do they correspond to in the original protein? What was the native peptide sequence (is in Arg or Lys in the Xxx modified position)?

 -Figure 3 A should show the value for individual serum (similar to the other figures) and not bars.

 -It would be good to always compare the reactivity to modified peptide to the responding native peptide as well.

 -What was the rational for selecting methylarginine as one of the investigated modifications?

 -What was the rational for focusing so much on the modification cross-reactivity to EBNA peptides in the RA samples? This has to be explained further. There is a large number of identified cit-peptides that are targeted in RA. Although EBNA may be interesting, a number of human peptides could also have been selected. How are the EBNA peptides selected, are these also know peptides in regular non-citrulline anti-EBNA responses? The paper would benefit from including different modified peptides originating from human proteins.

 -What are the details of the peptides investigated in figure 4 (residues in original protein etc)? Some seems to overlap with peptides in figure 3 but not all. Why is not homocitrulline included in this figure? Is 3c showing proteoglycan substitution (that is says in the legend?) or is it Arg(Me) that it says in the figure.

 -In figure 4a, there is some reactivity to the Arg(Me) peptide versions. Is that correlating with citrulline reactivity?

 -Why is not the pro-filaggrin peptide included in different modified forms in Figure 4?

 -How were the RA patients included in the study selected? What were the CCP titers, ACPA fine-specificity patterns?

 -In the discussion the authors discuss the reported antibody structure reported in Uysal et al. 2009. However, this antibody is a murine antibody and may not bind in the same way as a human ACPA. Since there is now a structure solved for a human ACPA mAb in complex with citrullinated peptides in Ge et al (PMID:30152126) that would make more sense to discuss in this context.

Author Response

Thank you very much for your comments. Your comments have indeed improved the final manuscript. They have been very constructive.

There has lately been an increasing interest of RA autoreactivity to other modified residues besides citrulline, including carbamylation, and the manuscript contributes to that literature and our understanding of ACPA. It is well written and focus on a very interesting topic, yet there are some questions to address that would substantially improve the manuscript:

 -It is overall difficult to follow what peptides that has been used in the different figures and what protein they originate from. In Figure 3 c, what proteins do the peptides originate from? What amino acid residues do they correspond to in the original protein? What was the native peptide sequence (is in Arg or Lys in the Xxx modified position)?

Answer: Table 1 has been added, which gives an overview of the peptides used in this study, which answers the questions raised by the reviewer. All modifications are introduced in Arg positions.

 -Figure 3 A should show the value for individual serum (similar to the other figures) and not bars.

Answer: Amended as requested. Antibody titres in figure 3A and 3B are illustrated for the individual serum and not bars. Moreover, the reactivity of healthy control sera been added as well.

 -It would be good to always compare the reactivity to modified peptide to the responding native peptide as well.

Answer: We agree that the correct controls are necessary. Arg-containing controls have been added where possible.

 -What was the rational for selecting methylarginine as one of the investigated modifications?

Answer: Citrullination arises as a posttranslational modification in relation to infection/inflammation and it may be regarded as a cellular response to an altered cellular state. Likewise, other Arg modifications could possibly arise in response to the altered cellular state, and methylation is a relatively common modification of Arg, which could play a role in the development of autoimmunity.

 -What was the rational for focusing so much on the modification cross-reactivity to EBNA peptides in the RA samples? This has to be explained further. There is a large number of identified cit-peptides that are targeted in RA. Although EBNA may be interesting, a number of human peptides could also have been selected. How are the EBNA peptides selected, are these also know peptides in regular non-citrulline anti-EBNA responses? The paper would benefit from including different modified peptides originating from human proteins.

Answer: Currently, no true autoantigen of RA has been identified, which is in accordance to that the ACPA response is very cross-reactive. EBV has been proposed to be involved in the onset of RA, hence ACPA reactivity to citrullinated human proteins, behave in a similar way. We have already characterized ACPA reactivity to several citrullinated human epitopes (Vimentin, pro-filaggrin, RA33). Our conclusion is that similar antibody reactivity pattern is observed, independent of the origin of the protein and peptide, as ACPA reactivity primarily depend on the central Cit-Gly motif in combination with (a random) peptide backbone for antibody reactivity.

As seen citrullinated peptides originating from human proteins are represented in all experiments conducted. Nevertheless, the antibody response to the viral citrullinated epitopes behaves in a similar way, which also is in accordance to previous findings. Collectively, it is in our believe that the EBV peptides are representative for ACPA reactivity to human peptides, which conforms to that recent findings indicate that ACPAs in fact are directed to a strain-specific EBNA2 antigen (Trier et al 2018, Antibodies to a strain-specific citrullinated epstein-barr virus peptide diagnoses rheumatoid arthritis. Sci Rep 2018, 8, 3684).     

The EBNA peptides are selected based on previous studies. Some of the peptides have been described as good ACPA candidates, whereas others have been described as poor candidates. This was done in other do determine whether addition of hCit would increase antibody reactivity and ultimately to determine whether antibody reactivity to citrullinated peptides and homocitrullinated peptides differ from each other. Based on current and previous findings, none of these peptides are targeted in regular EBV immune responses.

This discussion has been added to the discussion.  

-What are the details of the peptides investigated in figure 4 (residues in original protein etc)? Some seems to overlap with peptides in figure 3 but not all. Why is not homocitrulline included in this figure? Is 4c showing proteoglycan substitution (that is says in the legend?) or is it Arg(Me) that it says in the figure.

Answer: Peptide details are listed in table 1. Homocitrullinated peptides are not included as our main conclusion from the previous experiment is that they are poor ACPA substrates. The peptide used in figure 4c is originating from proteoglycan, containing the naturally occurring Arg-Gly motif, or substituted to Arg(me) or Cit-Gly. This has been clarified in the manuscript. This peptide has previously been identified as a potential ACPA substrate.

-In figure 4a, there is some reactivity to the Arg(Me) peptide versions. Is that correlating with citrulline reactivity?

 Answer: A correlation test was performed, which indicated a correlation of 0.6 (with max correlation=1), illustrating that no notable correlation was found between antibody reactivity to the Arg(me)-substituted peptides and the citrullinated peptides.

-Why is not the pro-filaggrin peptide included in different modified forms in Figure 4?

 Answer: We agree that it would have been interesting to test ACPA reactivity to the Arg(Me)-substituted profilaggrin as well. However, as concluded, none of the peptides tested experience significant antibody reactivity to the substituted versions compared to the controls, which with very high probability applies to the pro-filaggrin peptide as well.

-How were the RA patients included in the study selected? What were the CCP titers, ACPA fine-specificity patterns?

Answer: The RA patients were selected based on CCP reactivity. All RA patients were CCP2-positive, with antibody titres in the range of 25-3200 U/mL.

-In the discussion the authors discuss the reported antibody structure reported in Uysal et al. 2009. However, this antibody is a murine antibody and may not bind in the same way as a human ACPA. Since there is now a structure solved for a human ACPA mAb in complex with citrullinated peptides in Ge et al (PMID:30152126) that would make more sense to discuss in this context.

Answer: Amended as requested, the new crystal structure has been added to the discussion.

Reviewer 2 Report

In this manuscript, the authors have demonstrated that ACPAs from rheumatoid arthritis sera is specifically reactive to citrullinated epitopes. In the same experimental setting, the authors should also test the reactivity of sera obtained from healthy subjects, to citrullinated epitopes, which is a crucial control for this study. The authors should also provide additional information on RA patients. If they were CCP+/- and RF+/- and then group the reactivity of their sera accordingly.

Author Response

Amended as requested

Reactivity of healthy control sera was introduced in figure 2 and 3. Information of the RA sera was added to the materials and methods section. Only CCP2 positive sera were applied in this study, and only 3 sera were RF (IgM/IgA) negative. Based on the current patient information, no significant information would come from grouping the sera, as especially the CCP+/RF- group would be too small.   

Round  2

Reviewer 1 Report

Thank you for the improved revised manuscript and the clarifying comments! All questions have been appropriately addressed by the authors.